# Homology Modeling, Molecular Docking, Molecular Dynamic Simulation, and Drug-Likeness of the Modified Alpha-Mangostin against the β-Tubulin Protein of *Acanthamoeba Keratitis*

**DOI:** 10.3390/molecules27196338

**Published:** 2022-09-26

**Authors:** Tassanee Ongtanasup, Anisha Mazumder, Anupma Dwivedi, Komgrit Eawsakul

**Affiliations:** 1Department of Applied Thai Traditional Medicine, School of Medicine, Walailak University, Nakhon Si Thammarat 80160, Thailand; 2Faculty of Pharmacy, Charles University, 50005 Hradec Kralove, Czech Republic

**Keywords:** β-tubulin, *Acanthamoeba keratitis*, additive effect, synergistic effect, pharmacokinetics

## Abstract

Acanthamoeba species are capable of causing amoebic keratitis (AK). As a monotherapy, alpha-mangostin is effective for the treatment of AK; however, its bioavailability is quite poor. Moreover, the efficacy of therapy is contingent on the parasite and virulent strains. To improve readiness against AK, it is necessary to find other derivatives with accurate target identification. Beta-tubulin (BT) has been used as a target for anti-Acanthamoeba (*A. keratitis*). In this work, therefore, a model of the BT protein of *A. keratitis* was constructed by homology modeling utilizing the amino acid sequence from NCBI (GenBank: JQ417907.1). Ramachandran Plot was responsible for validating the protein PDB. The verified BT PDB was used for docking with the specified ligand. Based on an improved docking score compared to alpha-mangostin (AM), two modified compounds were identified: 1,6-dihydroxy-7-methoxy-2,8-bis(3-methylbut-2-en-1-yl)-9H-xanthen-9-one (C1) and 1,6-dihydroxy-2,8-bis(3-methylbut-2-en-1-yl)-9H-xanthen-9-one (C2). In addition, molecular dynamics simulations were conducted to analyze the interaction characteristics of the two bound BT–new compound complexes. During simulations, the TRP9, ARG50, VAL52, and GLN122 residues of BT-C1 that align to the identical residues in BT-AM generate consistent hydrogen bond interactions with 0–3 and 0–2. However, the BT-C2 complex has a different binding site, TYR 258, ILE 281, and SER 302, and can form more hydrogen bonds in the range 0–4. Therefore, this study reveals that C1 and C2 inhibit BT as an additive or synergistic effect; however, further in vitro and in vivo studies are needed.

## 1. Introduction

For many decades, humans have been continuously exposed to pathogenic Acanthamoeba [1,2]. The growing interest in traditional ethnomedicine might result in the identification of new medicinal molecules. Numerous plant species endowed with phytochemicals with potent anti-Acanthamoeba action have been established pharmacologically and therapeutically around the globe [3]. Modification of the active ingredient to make it more potent improves therapy [4,5]. Consequently, this study introduces a novel, computationally improved structured substance with enhanced inhibitory efficacy.

Tubulin is an important structural component of eukaryotic cells, where it plays a pivotal role in chromosomal segregation, organelle movement, and cellular motility [6]. Tubulin is involved in several cellular activities, such as mitosis, the active movement of proteins and organelles across the cytoplasm, and the preservation of cell shape [7].

The similarities between the amino acid sequences of alpha and beta tubulin are 40%, and both proteins have nearly identical three-dimensional (3D) structures [8]. A GTPase domain found in the self-polymerizing FtsZ protein family is shared by tubulins that are essential for the cell division of protozoa [9]. It will be interesting to determine the tubulin protein’s 3D structure given its functional importance, which in *Acanthamoeba keratitis* has yet to be resolved. Tubulin has thus been used as a target for antineoplastic [10], herbicide [11], antihelminthic [12], antifungal [13], and antiprotozoal [14] chemicals. Acanthamoeba is a ubiquitous opportunistic protozoan that is well known to cause severe human infections, including blinding keratitis and deadly encephalitis.

Mangosteen (*Garcinia mangostana* Linn.) (GML) is an Indian, Myanmar, Malaysian, Philippine, Sri Lankan, and Thai tropical tree. This slow-growing tree may reach heights of 6–25 m and has leathery, glabrous leaves [15]. The mangosteen fruits are reddish to almost dark purple in color, with a white, soft, juicy flesh that is somewhat acidic and sweet in taste and has a nice scent [16]. Mangosteen is among the most delicious tropical fruits and is hence regarded as “the queen of fruits”. Southeast Asians have utilized the pericarp of mangosteen fruit for generations to cure skin infections and wounds [15], amoebic dysentery [17], and other ailments. Mangosteen fruit pericarp is widely used in Ayurvedic medicine to treat inflammation and diarrhea [18], as well as cholera and dysentery [19]. Several secondary metabolites, including prenylated and oxygenated xanthones, have been discovered in mangosteen fruit [20]. The GML pericarp, fruit, bark, and leaves have all been shown to contain xanthones. A number of studies have shown that mangosteen-fruit-derived xanthones possess exceptional biological activity [21]. α,β,γ-Mangostins, garcinone E, 8-desoxygartanin, and gartanin are only some of the bioactive xanthones derived from mangosteen fruit [17].

Several investigations have shown that α-mangostin significantly inhibits Acanthamoeba. However, alpha mangostin lacks the specificity to block the tubulin of Acanthamoeba. because it induces phagocytic cells to eliminate intracellular organisms to demonstrate anticancer [22], antibacterial [17], antiviral [23], and antiprotozoal [24] activity. In addition, research indicates that it suppresses the release of histamine and the creation of prostaglandin E2, which is crucial for the prevention of allergic reactions [25]. Therefore, the study of the binding specificity of alpha mangostin to beta-tubulin is important for the development of a drug structure that is more beta-tubulin-specific, making the developed substance specific for the treatment of Acanthamoeba infection through an inhibition mechanism of beta-tubulin. This project develops the structure of alpha mangostin to selectively inhibit beta tubulin of Acanthamoeba.

## 2. Results

### 2.1. Beta-Tubulin (BT) Using Homology Modeling

BT’s 3D structure was created with SWISS-MODEL with a global model quality estimation (GMQE) score of 0.83. A GMQE score greater than 0.70 is regarded as a trustworthy predictor in general [26]. In addition, the crystal structure of BT was shown to have a similarity identity of 80% with template protein. The template protein used for modeling is Chlamydomonas reinhardtii (6u42) with a resolution of 3.4–3.6 Å [27]. The score of QMEANDisCo is the average per residue obtained by applying a distance restriction to QMEAN measurements [28]. In general, the QMEANDisCo score should not be below 0.6, indicating that the created protein is of poor quality [29]. For QMEANDisCo, the score of BT computed using the SWISS MODEL was 0.74 ± 0.05, suggesting that it is a high-quality predicted protein. Therefore, these results demonstrate the validity of the modeled BT’s quality. In Figure 1, ProCheck performed a Ramachandran plot analysis, revealing that 100 percent of residues fell inside the preferred region (89.1 percent) and the additionally allowed zone (10.9 percent), whereas no residues fell within the unsuitable region. As a result, the model has been frequently validated as being of high quality and has been applied to additional computational ligand–receptor interactions.

### 2.2. Pocket Binding Analysis

The CavityPlus web service has identified binding pockets for molecular docking [30,31]. This protein has been found to have nine cavities. As indicated in Table 1, two of these pockets had strong draggability, whereas seven had weak draggability. Strong binding sites were deemed superior to those with poor draggability, since this cavity displayed Prediction Maximum pkd values of 11.69 and 11.69 for ligand-targeted protein pocket binding sites. However, the remaining seven binding sites were unsuitable for compound binding. Figure 2 shows the presence of a strong cavity of BT on the purple surface. In addition, Table 1 lists the amino acid residues anticipated for all cavities. We consider that there is a significant likelihood of identifying possible molecular inhibitors by simply selecting the proper molecule for these binding sites. Therefore, these two sites are considered possibilities for compound binding design.

### 2.3. Molecular Docking

Both AutoDock and ArgusLab are used to compute molecular docking, and both programs are compared [32]. Homology-modeled beta-tubulin is derived from a Swiss model, and its energy consumption is lowered using the Swiss-Pdb Viewer and Chiron online web. The chemicals displayed in Table 2 are docked with BT; compared to AutoDock, the ArgusLab docking study has a higher binding score [32]. In autodock, the highest docking score was −10.56 kcal/mol for 1,6-dihydroxy-7-methoxy-2,8-bis(3-methylbut-2-en-1-yl)-9H-xanthen-9-one, and the lowest docking score was −10.18 kcal/mol for alpha mangostin. In ArgusLab, the highest binding score is −12.1683 kcal/mol for 1,6-dihydroxy-2,8-bis(3-methylbut-2-en-1-yl)-9H-xanthen-9-one, and the lowest binding score is −11.2156 kcal/mol for alpha mangostin. It has been demonstrated that the structurally modified compound inhibits BT more effectively than alpha-mangostin.

In addition, all substances were presented to capture targets in 2D and 3D images. In Figure 3, it was found that the position of 1,6-dihydroxy-7-methoxy-2,8-bis(3-methylbut-2-en-1-yl)-9H-xanthen-9-one (C1) is in the same position as alpha mangostin because of its identical amino acids, such as TRP 9, ALA 51, and VAL 52, corresponding to strong pocket binding position 1. It should be noted that the structurally modified compound C1 has a higher number of hydrogen bonds with BT at five locations compared to only two hydrogen bonding positions presented by alpha mangostin. Substances that have a modified structure, such as 1,6-dihydroxy-2,8-bis(3-methylbut-2-en-1-yl)-9H-xanthen-9-one C2, bind to BT with amino acids, including PHE254, MET255, GLY257, ALA259, TYR 258, ILE 281, and SER 302, where the handle matches the strong pocket binding position 2. Due to their different binding sites, the two substances can be used together to produce either additive or synergistic effects [30].

### 2.4. Quantum Chemical Calculations

Important hints on the stability of chemical compounds are provided by chemical reactivity descriptors such as EHOMO, ELUMO, ∆E (HOMO–LUMO energy gap), chemical hardness, electrophilicity, electron affinity, and electrophilicity. In this investigation, COSMOquick was used to determine chemical reactivity descriptors. The outcome demonstrated that Chemical hardness, softness, and the HOMO–LUMO energy gap are closely connected chemical characteristics. The maximum hardness principle [33] states that chemical hardness is a measure of chemical species’ stability. Less stable molecules have a small HOMO–LUMO energy gap, whereas more stable compounds have a high HOMO–LUMO energy gap. Softness is a measure of polarizability, and soft molecules readily donate electrons to a molecule or surface that accepts electrons [34]. Table 3 demonstrates that Alpha-mangostin had a small HOMO–LUMO energy gap, resulting in less stability than the structurally modified substance with a higher HOMO–LUMO energy gap, especially 1,6-dihydroxy-2,8-bis(3-methylbut-2-en-1-yl)-9H-xanthen-9-one, which had the highest HOMO–LUMO energy gap and the highest stability. Alpha-mangostin, with a hardness of 3.923, was found to be the smallest compared to the modified structure, which was more rigid and resulted in a high level of stability, which was found to be 1,6-dihydroxy-2,8-bis(3-methylbut-2-en-1-yl)-9H-xanthen-9-one. It is the most stable. However, it has been discovered that when hardness increases, electron affinity and electrophilicity decrease. The greater an electron affinity and electrophilicity, the less stable it is. The results indicated that Alpha-mangostin had the highest electron affinity and electrophilicity. This indicates that it was less stable than compounds with structural modifications.

### 2.5. Molecular Dynamics

As observed on the right side of Figure 4 in BT complexes with (A) alpha-mangostin, (B) 1,6-dihydroxy-7-methoxy-2,8–bis(3–methylbutyl)-9H–xanthen-9–one; C1 and (C) 1,6–dihydroxy-2,8–bis(3methylbutyl)-9H–xanthen-9–one; C2, the highest RMSD values were 0.55 nm 1 nm and 0.5 nm, respectively. In addition, it was discovered that the stability of C2 binding to BT increased after 10 ns. This was noticed as RMSD decreased and remained below 0.3 nm. Consequently, based on this finding, it can be inferred that the RMSD value of C2 was smaller than that of alpha-mangostin (positive control) during the simulated complexes, showing the stability of C2 with BT during the simulation time. This seems acceptable in light of the fact that C2 binds BT stably. According to Figure 4’s left side, the number of hydrogen bonds varies between 0 and 3 for (A) alpha-mangostin (positive control), the same as C2, and between 0 and 4 for C1. Based on these findings, it was determined that C1 can create more hydrogen bonds during MD. However, RMSD was relatively high compared to both compounds, indicating that C1 binding to BT was unstable despite the formation of up to four hydrogen bonds, whereas C2 was found to be stable due to a lower RMSD value, indicating a more stable C2-BT binding complex through the formation of up to three hydrogen bonds.

### 2.6. Synthesis and Structural Analysis via NMR

Both structurally modified substances 1,6-dihydroxy-7-methoxy-2,8-bis(3-methylbut-2-en-1-yl)-9H-xanthen-9-one; C1 and 1,6-dihydroxy-2,8-bis (3-methylbut-2-en-1-yl)-9H-xanthen-9-one; C2 were synthesized through the use of IBM RXN chemistry with artificial intelligence (AI). The results showed that C1 could be synthesized via O-MOM deprotection [35], as shown in Figure 5A, which is a reaction among water, hydrochloric acid, and 1-hydroxy-7-methoxy-6-(methoxymethoxy)-4a,9a-dihydro-2,8-bis(3-methylbut- 2-en-1-yl)-9H-xanthen-9-one. However, C2 can be synthesized through three steps, starting with (1) a Sandmeyer reaction [36]^,^ as shown in Figure 5B, through the reaction of 6-amino-1-hydroxy-4a,9a-dihydro-2,8-bis(3-methylbut-2-en-1-yl)-9H-xanthen-9-one, hydrochloric acid, water, sodium nitrite, and copper(I) bromide, which produces 6-bromo-1-hydroxy-4a,9a-dihydro-2,8-bis(3-methylbut-2-en-1-yl)-9H-xanthen-9-one. (2) Miyaura borylation reaction [37]^,^ as shown in Figure 5C, undergoes the reaction of the substances from reaction 1, dioxane, potassium acetate, bis(pinacolato)diboron, palladium(II) chloride, and 1,1′-. Bis(diphenylphosphino) ferrocene produces 1-hydroxy-6-(4,4,5,5-tetramethyl-1,3,2-dioxaborinan-2-yl)-4a,9a-dihydro-2,8-bis. (3-Methylbut-2-en-1-yl)-9H-xanthen-9-one 3) Borono to hydroxy reaction [38] as shown in Figure 5D through the reaction of the substances obtained from reaction 2, THF, hydrogen peroxide, and sodium hydroxide to form C2.

After the synthesis of C1 and C2, these substances were subjected to NMR analysis, and proton peaks were predicted, as shown in Figure 6. Alpha-mangostin has the following ^1^H NMR (500 MHz) peaks: δ = 1.72–1.73 ppm (quartet, CH_3_), 3.83 (singlet, OCH_3_), 2.99–4.09 (doublet, CH), 5.06–5.21 (multiplet, CH), 5.41 (doublet, CH), 5.53 (quartet, CH), 6.34 (singlet, CH), 7.67–14.00 (singlet, OH). The removal of the OH group from the structure of Alpha-mangostin becoming C1 causes the peak to shift from 8.12 ppm to 5.99 ppm, resulting in the following ^1^H NMR (500 MHz) peaks: δ = 1.72–1.73 ppm (quartet, CH_3_), 2.86–3.50 (doublet, CH), 3.83 (singlet, OCH_3_), 3.95 (doublet, CH), 5.06–5.21 (multiplet, CH), 5.37(octet, CH), 5.56–5.99 (quartet, CH), 6.34(singlet, CH), 7.67–14.00 (singlet, OH). The removal of the OH and OCH3 groups from the Alpha-mangostin structure causethe s C2 structure. This caused two significant peak shifts, from 8.12 to 5.99 ppm and 3.83 to 6.48 ppm, resulting in the following C2 peaks in ^1^H NMR (500 MHz): δ = 1.72–1.73 ppm (quartet, CH_3_), 2.86–3.22 (doublet, CH), 3.26–3.32 (quartet, CH), 3.94 (doublet, CH), 5.15–5.21 (multiplet, CH), 5.31(octet, CH), 5.57–5.98 (quartet, CH), 6.27(doublet, CH), 6.48 (sextet, CH), 8.18–14.00 (singlet, OH).

### 2.7. Physicochemical and Pharmacokinetic Analysis of the Compounds

The findings of the projected gastrointestinal absorption (GIA) of the selected substances are presented in Table 4. All the substances showed a high absorption probability in the gastrointestinal system. This indicates that these chemicals may be absorbed in the gastrointestinal system following oral dosing [39]. The BBB is the layer of brain microvascular endothelial cells that separates the brain from the blood [40]. The capacity of the compounds to cross the BBB was investigated, and the findings are displayed in Table 4. According to the data, none of the chemicals exhibited the ability to cross the BBB, which is advantageous since it reduces the possibility that they may cause detrimental effects on the CNS [41]. P-glycoprotein (P-gp) is a membrane transporter of intracellular and extracellular substances [42]. According to estimates, only modified compounds (C1 and C2) are nonsubstrates for P-gp. This suggests that the compounds would not be impacted by the efflux activity of P-gp, which removes chemicals from cells, resulting in therapeutic failure due to lower than desired amounts. Thus, C1 and C2 have the potential to resist various targets [43].

Prediction of the metabolism of lead compounds is a top priority during the drug development procedure [39]. Table 4 displays the findings of the drugs’ predicted metabolism against five isoforms of the cytochrome P450 (CYP) monooxygenase family: CYP1A2, CYP2C19, CYP2C9, CYP2D6 and CYP3A4. None of the substances inhibited CYP1A2 and CYP2D6, whereas only C2 inhibited CYP2C19. In biological systems, cytochrome P450 monooxygenase is essential for drug metabolization and removal. The noninhibitory effect of the discovered compounds on these enzymes indicates that they have a high probability of being converted and hence be bioavailable following oral administration [44]. On the other hand, the inhibition of CYP isomers by the substances might result in poor bioavailability due to their inability to be metabolized and severe side effects due to compound accumulation [45,46,47].

From the results mentioned above, it can be concluded that alpha-mangostin and C1 are likely to have fewer side effects than C2, while both C1 and C2 are highly therapeutic because they would not be affected by the efflux action of P-gp. All three compounds are well absorbed through oral administration. C1 provides good pharmacokinetic value in both bioavailability and reduces the side effects, while C2 can also have good bioavailability, but there may be side effects of using a high dose of C2, so it is necessary to control the amount of C2 appropriately while using it to prevent side effects.

The drug-likeness of compounds was assessed based on their physicochemical features in order to find drug candidates. Rule-based filters were classified into the three groups listed below.

(1) Lipinski’s filter [48] takes into account the following variables: the molecular weight is less than or equal to 500 and the number of hydrogen bond acceptors is less than 10, and the number of hydrogen bond donors is less than 5, and MLOGP (lipophilicity) of less than 4.15 then the molecule meets the criteria.

(2) Veber’s filter [49] incorporates the following settings: Veber’s filter includes only those molecules with a total polar surface area of less than or equal to 140 and a number of rotatable bonds of less than or equal to 10.

(3) Egan’s filter [50] takes into account the following criteria: a total polar surface area of less than or equal to 131.6 and WLOGP (hydrophilic) of less than or equal to 5.88.

Based on the results of the investigations, all three compounds were found to be suitable for drug candidates, as observed in Table 5. They have an optimal structure for development as a treatment for *Acanthamoeba keratitis*.

## 3. Discussion

Alpha-mangostin used to treat *Acanthamoeba keratitis* (AK) is poorly bioavailable. Therefore, there is an urgent need for medicines that can effectively treat AK with minimal side effects. Design, synthesis, and pharmacological evaluation are followed by an examination of the drug’s safety, all of which require a significant investment of time, money, and manpower. Failure to study any of the aforementioned parameters renders the chemical unfit for the intended therapeutic use. In light of this perspective and the improvement of computational approaches, the present study attempts to reduce the aforementioned concerns by utilizing in silico molecular modeling investigations, molecular docking, molecular dynamics, and computational toxicity assessments.

Molecular docking techniques try to anticipate the optimal mode of binding between a ligand and a macromolecular partner. Molecular dynamics (MD) is a computer approach that mimics the dynamic behavior of molecular systems as a function of time. Therefore, such procedures require a macromolecule as a receptor and a ligand. Both procedures are important and relevant methods for drug development. If a substance can form a large number of hydrogen bonds to the target protein by molecular docking studies, it results in greater stability when studied with molecular dynamics. However, confirmation studies are needed. This study found that both modified compounds bind to the target protein better than alpha mangostin, which is observed in the 4–5 hydrogen bonds, whereas alpha mangostin can only be formed at two hydrogen bonds, which means that both compounds bind more easily to beta-tubulin. It corresponds to the effect of stability. The stability of binding through molecular dynamic results showed C2 has the highest stability because RMSD has the lowest value. The process of generating two novel compounds begins with the discovery that TRP 9 on the alpha-mangostin structure had an incompatible interaction with BT. Therefore, the structure was enhanced by eliminating an undesirable structure that was incompatible with binding. There are two new chemicals formed: 1,6-dihydroxy-7-methoxy-2,8-bis(3-methylbut-2-en-1-yl)-9H-xanthen-9-one; C1 and 1,6-dihydroxy-2,8-bis(3-methylbut-2-en-1-yl)-9H-xanthen-9-one; C2. After the study, both compounds were able to attach to BT more effectively. C1 may create hydrogen bonds at a maximum of five sites, TRP 9, ARG 50, VAL 52, and GLN 122, while C2 can form hydrogen bonds at a maximum of four sites, ILE 281, TYR 258, and SER 302, compared to Alpha mangostin, which can form just two hydrogen bonds. In addition, binding stability was investigated using molecular dynamics. The RMSD between C2 and BT is less than 0.3 nm indicating high stable binding. However, studies have shown that C2 inhibits three isoforms as CYP2C9, CYP2C19, and CYP3A4, resulting in decreasing drug metabolizing effect and is a major contributor to the adverse effects of the C2. Therefore, it is very important to control the use of C2 to prevent the side effects of C2 use. While C1 may bind to BT in an unstable manner. However, C1 has fewer side effects due to its use. Such agents are unable to inhibit CYPs, resulting in better elimination of the C1 from the body. If considering the advantages and disadvantages of both substances, the combination of the two substances will help to maximize the benefit of the treatment. This study discovered that both changed structures may bind to BT in distinct sites, resulting in additive or synergistic effects when both compounds are used together. However, current hypotheses on this modified alpha-mangostin require more in vivo and in vitro research to determine the optimal therapeutic effectiveness and lowest toxicity.

The morbidity and mortality associated with *Acanthamoeba keratitis* have not significantly decreased over the past many decades [51]. The development of antiacanthamoebic chemicals has not resulted in effective chemotherapeutics on the market. The rate at which innovative antiacanthamoebic chemotherapies with translational value have been developed and the pharmaceutical industry’s lack of interest in developing such chemotherapies have been disappointing. Alternatively, the market for contact lenses/contact lens disinfectants is a multibillion-dollar sector that has been successful and profitable. Greater knowledge of structurally modified compounds, β-tubulin protein, and mechanisms of action will aid in the creation of more effective chemotherapies against *Acanthamoeba keratitis*. This research opens up a new method for structural improvement of the known compound through the removal of the non-binding portion of the target protein. This method can be applied to other drug developments in the future.

## 4. Materials and Methods

### 4.1. Evaluation and Homology Modeling of the Beta-Tubulin Protein of Acanthamoeba Keratitis

The beta-tubulin protein of *Acanthamoeba keratitis* was modeled using the SWISS-MODEL server (https://swissmodel.expasy.org/), and the model quality was evaluated using Qualitative Model Energy Analysis (QMEAN) (https://swissmodel.expasy.org/qmean). SWISS-MODEL is a completely automated tool used to estimate the three-dimensional structure of proteins. Homology modeling approaches build 3D models [52]. SWISS-MODEL was fed the FASTA format of beta tubulin protein from the NCBI database (http://www.ncbi.nlm.nih.gov) as stated in Table 6. The projected model of beta tubulin from SWISS-MODEL was an input for the QMEAN study. The QMEAN server offers access to QMEANDisco [28]. It calculates the quality of protein structure prediction models [53]. The minimized energy of beta-tubulin was then conducted using Swiss-Pdb Viewer [54] and the Chiron online web server (https://dokhlab.med.psu.edu/chiron/login.php) [55,56] to ensure appropriate docking [57]. Energy minimization was performed to minimize the protein’s potential energy. ProCheck was used to verify the anticipated three-dimensional structures by creating the Ramachandran plot [58]. In addition, the Cavity Plus server [59] was also used to locate the binding pockets.

### 4.2. Screening of Compounds Using Arguslab and Autodock

Before molecular docking, compounds were illustrated using Pubchem sketcher V2.4 [60]. Avogadro version 1.1.0 [61] was used to construct three-dimensional models of the compounds, following which they were optimized in Avogadro and energy minimized in ArgusLab [62]. The molecular docking was perform using alpha mangosteen and modified alpha mangosteen compounds (1,6-dihydroxy-7-methoxy-2,8-bis(3-methylbut-2-en-1-yl)-9H-xanthen-9-one; C1 and 1,6-dihydroxy-2,8-bis(3-methylbut-2-en-1-yl)-9H-xanthen-9-one; C2), which were prepared using ArgusLab and Autodock [63] for the docking method, the protein was set as a rigid molecule and the ligands as flexible. The preparation of proteins prior to molecular docking via the web server of https://www.playmolecule.com/proteinPrepare/. Briefly, the protein includes the titration of the protonation states using *PROPKA 3.1* and addition of missing atoms, and overall optimization of the H- network using *PDB2PQR 2.1*. Then, the specific pocket binding that was previously positioned from the Cavity Plus server led to the specified grid box with xyz points. It was set at a size of 106 × 82 × 112, with the grid position at 296.419, 243.257, and 405.536 and spacing at. 0.375 Å. For optimal conformations, the lowest binding energy (kcal/mol) was used for analyzing the results. The study of complicated protein-ligand structure 2D and 3D interactions, including varied types and numbers of linkages, was carried out using Biovia Discovery Studio 2020 Visualizer [64].

### 4.3. Quantum Chemical Calculations

Quantum chemical parameters, including electron affinity, chemical potential, hardness, electrophilicity, highest occupied molecular orbital (HOMO), and lowest unoccupied molecular orbital (LUMO), were submitted to COSMOquick based on an alternative method of quantum chemical calculations. Approximated quantum chemical parameters were created directly from the specified SMILES strings using a library of about 200,000 pre-computed molecules using the COSMOquick 1.7 program [65] for all compounds.

### 4.4. Molecular Dynamics Modeling

The molecular dynamics were simulated for 50 ns by implying GROMOS96 43a1 forcefield with GROMACS (v5.1.2) package [66] using WebGro server. Using the online WEBGRO Macromolecular Simulations service (https://simlab.uams.edu/ProteinWithLigand/protein with ligand.html) [67,68], molecular dynamics simulations of beta-tubulin protein in the presence of alpha-mangosteen, C1, and C2 were conducted. Prior to running MD simulations, the topology of alpha-mangosteen, C1, and C2 must be generated using the GlycoBioChem PRODRG2 service (http://davapc1.bioch.dundee.ac.uk/cgi-bin/prodrg/submit.html) [69]. It is a web-based server. The force field used for MD simulations is GORMACS96 43a1 for beta-tubulin with these chemicals, the SPS water model in cubic box form, and sodium chloride. Steepest descent integrators were then used every 5000 steps to lower the complex’s energy. The NVT/NPT equilibration was conducted at 300 K and 1 bar of pressure. Due to a limitation, the MD integrator utilized was Leap-frog for a simulation length of 50 ns, and the number of frames per MD simulation was fixed at 1000 [70]. To determine the stability of a complex, MD simulations provided numerous trajectories, such as the root-mean-square deviation (RMSD) and hydrogen bonds (HBs).

### 4.5. Predicting Chemical Shifts in NMR Based on Knowledge of the Structure

The accurate chemical shift predictions of H NMR were attained by computational analysis. ChemAxon Reactor 22.13.0 software [71] was used to create the structures of three compounds for NMR peak prediction, initially by writing the 2D structure based on Table 1, followed by calculations > NMR > HNMR prediction.

### 4.6. Physicochemical and Pharmacokinetic Analysis of the Compounds Using the SwissADME Tool

As described in Section 4.2, the chemical structures of compounds were created. SwissADME web online tool [39] was employed to predict pharmacokinetics, SMILES format was added in the upper left corner, and then ADME computation was performed.

## Figures and Tables

**Figure 1 molecules-27-06338-f001:**
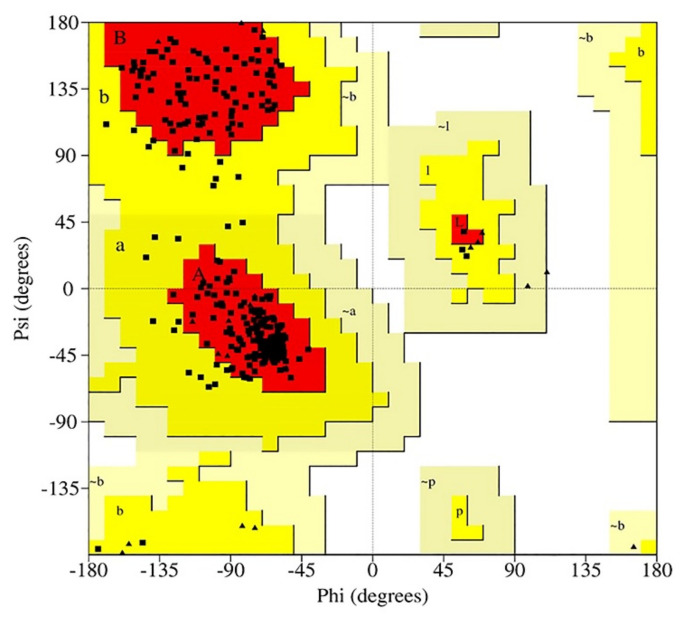
The ramachandran figure illustrates the phi-psi torsion angles for each beta-tubulin residue. The red regions represent the most desirable phi-psi value combinations. The white region represents an undesirable phi-psi combination.

**Figure 2 molecules-27-06338-f002:**
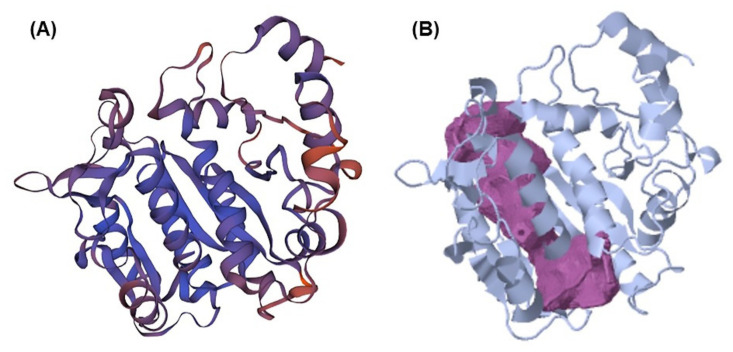
(**A**) Homology model of beta-tubulin by Swiss Model. (**B**) The detected cavity with the strong-binding site of beta-tubulin.

**Figure 3 molecules-27-06338-f003:**
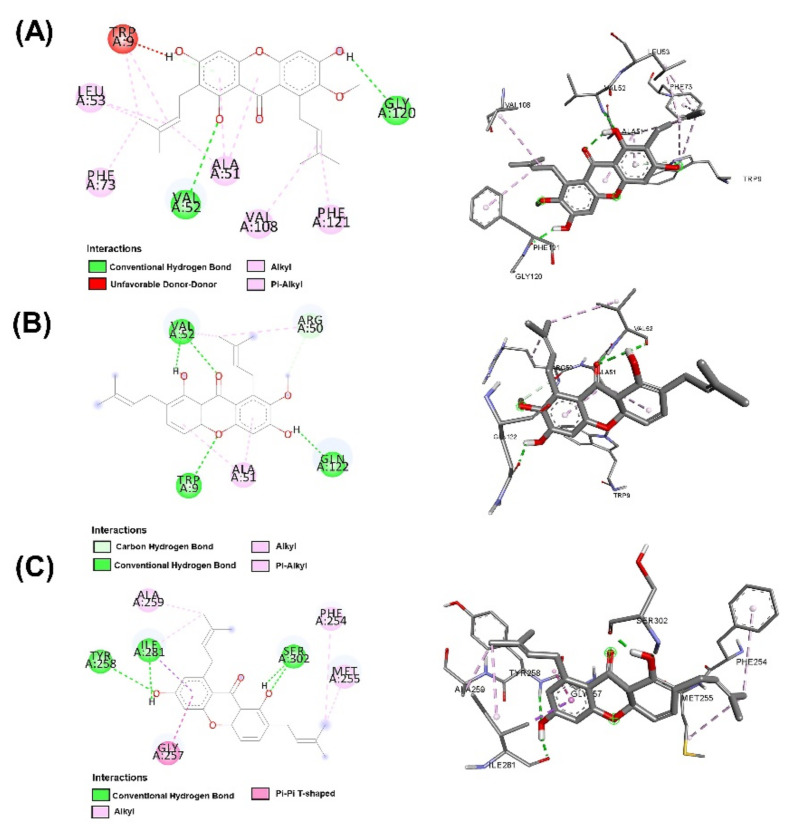
Post docking analysis visualized by Discovery Studio visualizer in 2D and 3D poses in beta-tubulin with (**A**) alpha-mangostin, (**B**) 1,6-dihydroxy-7-methoxy-2,8-bis(3-methylbut-2-en-1-yl)-9H-xanthen-9-one, and (**C**) 1,6-dihydroxy-2,8-bis(3-methylbut-2-en-1-yl)-9H-xanthen-9-one.

**Figure 4 molecules-27-06338-f004:**
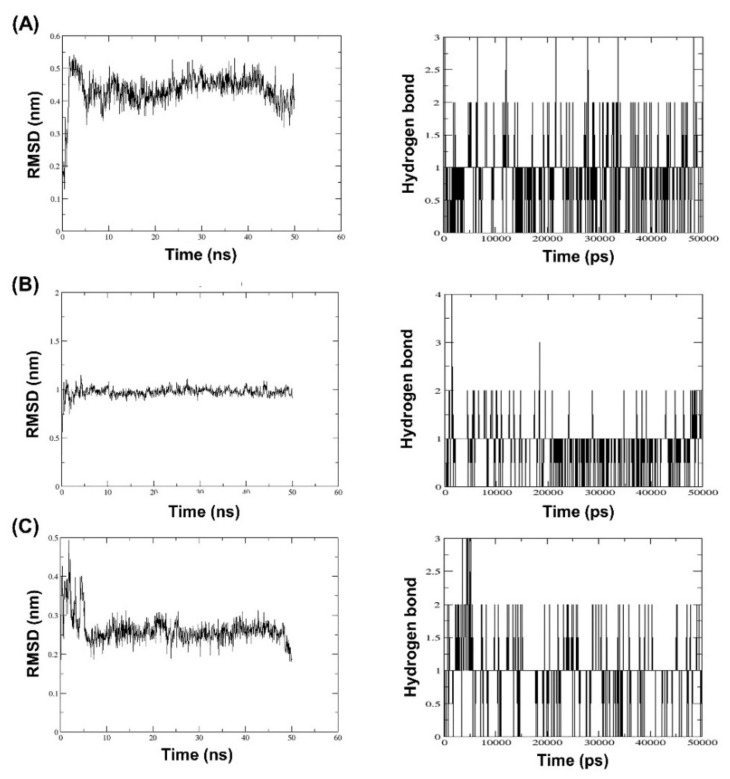
Molecular dynamics simulation of beta-tubulin with (**A**) alpha-mangostin, (**B**) 1,6-dihydroxy-7-methoxy-2,8-bis(3-methylbut-2-en-1-yl)-9H-xanthen-9-one, and (**C**) 1,6-dihydroxy-2,8-bis(3-methylbut-2-en-1-yl)-9H-xanthen-9-one.

**Figure 5 molecules-27-06338-f005:**
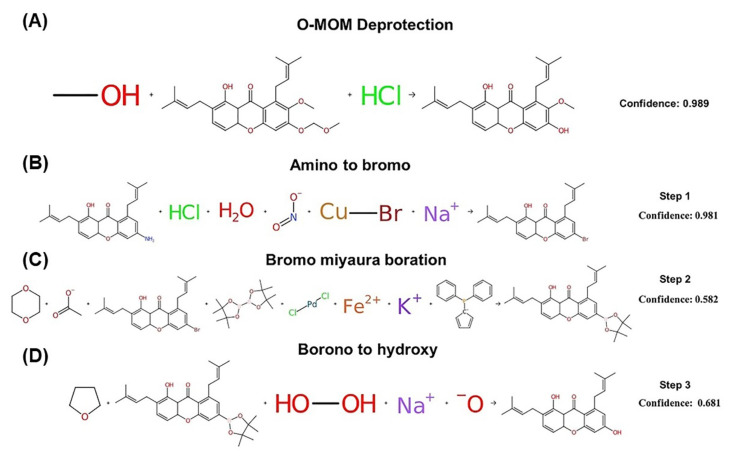
Planning synthesis of compounds: (**A**) 1,6-dihydroxy-7-methoxy-2,8-bis(3-methylbut-2-en-1-yl)-9H-xanthen-9-one, (**B**) 6-bromo-1-hydroxy-4a,9a-dihydro-2,8-bis(3-methylbut-2-en-1-yl)-9H-xanthen-9-one, (**C**) 1-hydroxy-6-(4,4,5,5-tetramethyl-1,3,2-dioxaborinan-2-yl)-4a,9a-dihydro-2,8-bis. (3-Methylbut-2-en-1-yl)-9H-xanthen-9-one, and (**D**) 1,6-dihydroxy-2,8-bis (3-methylbut-2-en-1-yl)-9H-xanthen-9-one by IBM RXN chemistry.

**Figure 6 molecules-27-06338-f006:**
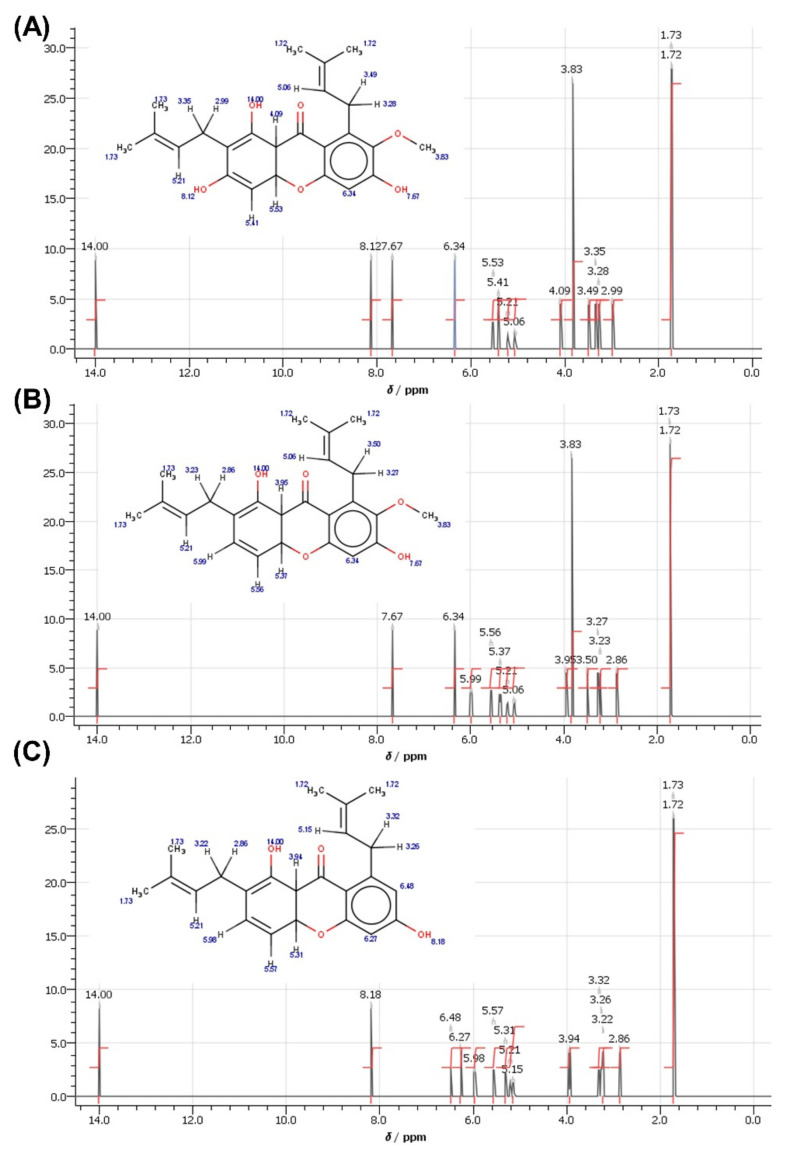
The predicted 1H NMR spectra for (**A**) alpha-mangostin, (**B**) 1,6-dihydroxy-7-methoxy-2,8-bis(3-methylbut-2-en-1-yl)-9H-xanthen-9-one and (**C**) 1,6-dihydroxy-2,8-bis (3-methylbut-2-en-1-yl)-9H-xanthen-9-one.

**Table 1 molecules-27-06338-t001:** Amino sequences of beta-tubulin in each pocket site.

No.	Residues	Druggability	Pred.Max pKd
**1**	GLY1, ASN2, GLN3, ILE4, GLY5, LYS6, LYS7, PHE8, TRP9, ASP33, ARG34, ILE35, ASN36, VAL37, TYR38, PHE39, THR40, GLU41, PRO49, ARG50, ALA51, VAL52, LEU53, VAL54, ASP55, LEU56, GLU57, PRO58, GLY59, THR60, MET61, ILE64, PHE73, PHE78, GLY84, ALA85, GLY86, ASN87, ASN88, VAL104, VAL107, VAL108, ARG109, LYS110, GLU111, ALA112, GLU113, ASN114, SER115, ASP116, LEU117, LEU118, GLN119, GLY120, PHE121, GLN122, VAL123, CYS124, HIS125, SER126, LEU127, GLY128, GLY129, GLY130, THR131, GLY132, SER133, GLY134, MET135, GLY136, THR137, LEU139, ILE140, ILE143, PHE147, ARG150, MET151, MET152, CYS153, PHE155, VAL157, MET158, PRO159, ASP165, THR166, GLU169, ASN172, ASN192, LEU195, TYR210, LEU213, ASN214, VAL217, MET221, VAL224, THR225, SER227, LEU228, ARG229, PHE230, SER236, ASP237, LEU238, ARG239	Strong	11.69
**2**	GLU10, VAL11, ILE12, ASP14, GLU15, MET151, CYS153, PHE155, MET158, VAL181, GLN186, VAL187, MET188, CYS189, ILE190, HIS215, SER218, GLN219, VAL220, MET221, SER222, GLY223, VAL224, THR225, ALA226, ARG229, PHE230, PRO231, LEU234, SER236, ASP237, LEU238, ARG239, LYS240, LEU241, ALA242, VAL243, ASN244, LEU245, ILE246, PRO247, PHE248, ARG250, LEU251, HIS252, PHE253, PHE254, MET255, VAL256, GLY257, TYR258, ALA259, PRO260, LEU261, THR262, ARG270, ASN271, PHE272, ASN273, VAL274, ALA275, GLU276, ILE277, THR278, GLN279, GLN280, ILE281, PHE282, ASP283, ALA284, ASN286, ILE287, MET288, ALA289, ALA290, CYS291, ASP292, PRO293, ARG294, HIS295, GLY296, ARG297, TYR298, LEU299, THR300, ALA301, SER302, ALA303, VAL304, PHE305, ARG306, GLY307, LYS308, VAL309, GLU313, VAL314, ASP315, GLN316, GLN317, MET318, LEU319, ASN320	Strong	11.61
**3**	ARG144, PRO148, ASP149, ARG150, MET151, GLN179, LEU180, VAL181, GLU182, ASN183, ALA184, ASP185, GLN186, LEU238, ARG239, LYS240, LEU241, ALA242, VAL243, ASN244, LEU245, ILE246, PRO247, PHE248, PRO249, ARG250, LEU251, HIS252	Weak	11.48
**4**	ASP14, GLU15, HIS16, ASP27, ASP28, PRO29, LEU30, GLN31, LEU32, GLN219, GLY223, ALA226, SER227, LEU228, ARG229, PHE230, PRO231, GLY232, GLN233, LEU234, ASN235, SER236, ASP237, LYS240, LEU241, ASN244, LEU245, TYR258, ALA259, PRO260, PHE282, THR300, ALA301, SER302, ALA303, VAL304, PHE305, ARG306, GLY307, LYS308, VAL309, SER310, THR311, LYS312, VAL314, ASP315, GLN316, MET318	Weak	11.31
**5**	ASN2, GLN3, ILE4, LYS6, LYS7, GLU10, GLY59, THR60, MET61, ASP62, ALA63, ILE64, ARG65, SER66, GLY67, VAL68, ASN209, TYR210, SER211, ASP212, ASN214, HIS215	Weak	8.39
**6**	SER160, PRO161, LYS162, ASP191, ASN192, GLU193, ALA194, LEU195, TYR196, ASP197, ILE198, ARG201, ASP283, ALA284, LYS285, ASN286, ILE287, MET288, ALA289, ALA290, CYS291, ASP292, PRO293, ARG294, HIS295	Weak	7.32
**7**	ASP55, LEU56, PRO58, VAL79, PHE80, GLY81, GLN82, SER83, GLY84, ALA85, LYS91, THR95, GLU96, GLY97, GLU99, LEU100, VAL101, SER103, MET135	Weak	7.29
**8**	ASP197, PHE200, ARG201, THR202, LEU203, LYS204, PRO260, LEU261, THR262, ALA263, PRO264, ASN265, SER266, THR267, TYR269, ARG270, ASN271, GLU276, GLN279, GLN280, ILE281, PHE282, ASP283, ALA284, LYS285, ASN286	Weak	6.73
**9**	ALA85, GLY86, ASN87, ASN88, TRP89, ALA90, LYS91, TYR94, THR166A, VAL167, VAL168, PRO170, TYR171, ASN172, THR174, LEU175	Weak	6.43

**Table 2 molecules-27-06338-t002:** The binding energy of compounds binding to beta-tubulin analyzed by ArgusLab and AutoDock tool.

Compounds	Smile	IUPAC Name	Binding Energy (kcal/mol)		Inhibition Constant, Ki
			Arguslab	Autodock	
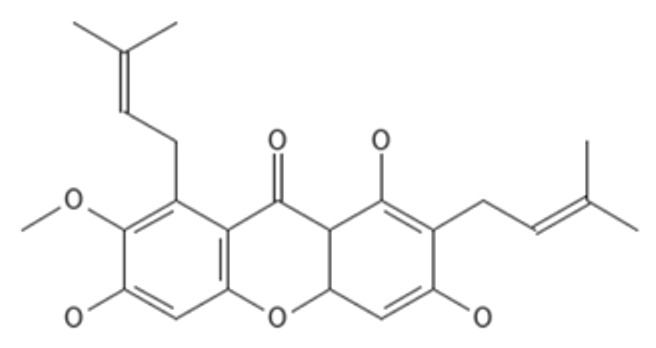	C1(=C(C(=CC3=C1C(C2C(C=C(C(=C2O)CC=C(C)C)O)O3)=O)O)OC)CC=C(C)C	1,3,6-Trihydroxy-7-methoxy-2,8-bis(3-methylbut-2-en-1-yl)-9H-xanthen-9-one	−11.22	−10.18	34.35 nM
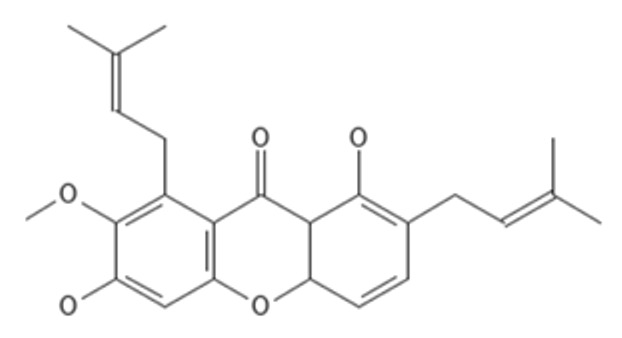	C1(=C(C(=CC3=C1C(C2C(C=CC(=C2O)CC=C(C)C)O3)=O)O)OC)CC=C(C)C	1,6-dihydroxy-7-methoxy-2,8-bis(3-methylbut-2-en-1-yl)-9H-xanthen-9-one	−11.81	−10.56	18.02 nM
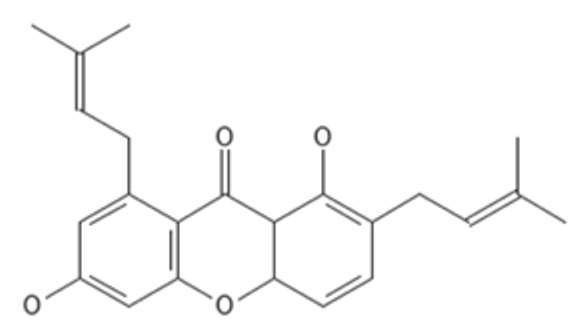	C1(=CC(=CC3=C1C(C2C(C=CC(=C2O)CC=C(C)C)O3)=O)O)CC=C(C)C	1,6-dihydroxy-2,8-bis(3-methylbut-2-en-1-yl)-9H-xanthen-9-one	−12.17	−10.52	19.43 nM

**Table 3 molecules-27-06338-t003:** Calculated quantum chemical parameters of alpha-mangostin and modified alpha-mangostin.

Quantum Chemistry Parameters	Alpha-Mangostin	1,6-dihydroxy-7-methoxy-2,8-bis(3-methylbut-2-en-1-yl)-9H-xanthen-9-one	1,6-dihydroxy-2,8-bis(3-methylbut-2-en-1-yl)-9H-xanthen-9-one
**Electron affinity**	0.951	0.723	0.653
**Chemical potential**	−4.874	−4.807	−4.739
**Hardness**	3.923	4.084	4.086
**Electrophilicity**	3.028	2.829	2.748
**HOMO**	−8.797	−8.890	−8.825
**LUMO**	−0.951	−0.723	−0.653

**Table 4 molecules-27-06338-t004:** Physicochemical and Pharmacokinetics of compounds.

Properties	Alpha-Mangostin	1,6-dihydroxy-7-methoxy-2,8-bis(3-methylbut-2-en-1-yl)-9H-xanthen-9-one	1,6-dihydroxy-2,8-bis(3-methylbut-2-en-1-yl)-9H-xanthen-9-one
**Physicochemical Properties**			
**Formula**	C_24_H_28_O_6_	C_24_H_28_O_5_	C_23_H_26_O_4_
**Molecular weight**	412.48 g/mol	396.48 g/mol	366.45 g/mol
**Num. H-bond acceptors**	6	5	4
**Num. H-bond donors**	3	2	2
**Molar Refractivity**	116.12	114.55	108.06
**Pharmacokinetics**			
**GI absorption**	High	High	High
**BBB permeant**	No	No	No
**P-gp substrate**	Yes	No	No
**CYP1A2 inhibitor**	No	No	No
**CYP2C19 inhibitor**	No	No	Yes
**CYP2C9 inhibitor**	Yes	Yes	Yes
**CYP2D6 inhibitor**	No	No	No
**CYP3A4 inhibitor**	Yes	Yes	Yes

**Table 5 molecules-27-06338-t005:** Drug likeness of Alpha-mangostin and modified compounds of Alpha-mangostin.

Compounds	Lipinski	Veber	Egan
**Alpha-mangostin**	Yes; 0 violation	Yes	Yes
**1,6-dihydroxy-7-methoxy-2,8-bis(3-methylbut-2-en-1-yl)-9H-xanthen-9-one**	Yes; 0 violation	Yes	Yes
**1,6-dihydroxy-2,8-bis(3-methylbut-2-en-1-yl)-9H-xanthen-9-one**	Yes; 0 violation	Yes	Yes

**Table 6 molecules-27-06338-t006:** Fasta format of the beta-tubulin sequence from NCBI data.

Targeted Protein	Protein Sequences
Beta-Tubulin *(Acanthamoeba keratitis)*	GNQIGKKFWEVIADEHGIDGTGKYIGDDPLQLDRINVYFTEASGGNYVPRAVLVDLEPGTMDAIRSGVHGKLFRPDNFVFGQSGAGNNWAKGHYTEGAELVDSVLDVVRKEAENSDLLQGFQVCHSLGGGTGSGMGTLLISKIREEFPDRMMCTFSVMPSPKVSDTVVEPYN ATLSVHQLVENADQVMCIDNEALYDICFRTLKLSNPNYSDLNHLVSQVMSGVTASLR FPGQLNSDLRKLAVNLIPFPRLHFFMVGYAPLTAPNSTAYRNFNVAEITQQIFDAKNIM AACDPRHGRYLTASAVFRGKVSTKEVDQQMLN

## Data Availability

All data, tables, and figures are original.

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
