# Peer review of "Homology Modeling, Molecular Docking, Molecular Dynamic Simulation, and Drug-Likeness of the Modified Alpha-Mangostin against the β-Tubulin Protein of Acanthamoeba Keratitis"

_molecules, 2022, doi:10.3390/molecules27196338_

Round 1

Reviewer 1 Report

Comments

1. Inform the versions and references for the software used in molecular docking studies.

2. The legend of Figure 3, needs to be improved, on the left side is 3D and on the right side is the 2D diagram, inform the interactions of the 2D diagram (colors).

3. I could not find the methodology used for the molecular docking studies, grid size and region and other information.

4. Was the protein protonated? if yes, inform how the protonation was performed, if not, redo the docking studies with the protonated protein.

5. In section 2.6, an important property to be analyzed would be the lipinski rule.

6. I did not find the methodology used for the molecular dynamics simulations, software, version, properties, simulation time, binding energy of the complexes and other characteristics.

Author Response

Please consider our revised manuscript ‘Homology modeling, molecular docking, molecular dynamic simulation,and drug-likeness of the modified alpha-mangostin against the β-tubulin protein of Acanthamoeba keratitis’. We would like to thank the reviewer for the time invested in reviewing this manuscript and for the valuable comments provided. We have considered the comments raised by the reviewers, and the responses detailed in the attached document have been implemented in the revised manuscript and highlighted with yellow colour.

Reviewer 2 Report

Major revision is required

1. All figures should be replaced with high-resolution figures and text should be readable. 

2. The statement in the introduction: structure of alpha mangostin to selectively inhibit beta-tubulin 71 of Acanthamoeba should be re-written and the rationale of the study should be clearly incorporated during revision.

3. The binding energies and the inhibition values should be correlated and also discuss the type of inhibition shown by the compounds

3. Which PDB id was used as a template in modeling and what was the resolution

4. The docking results and simulation data should be more correlated in the discussion part

5. Any stability studies including quantum chemistry parameters were observed?

6. Discussion should be increased at the end according to this journal perspective also and how this work will open new ways for β-tubulin protein of Acanthamoeba keratitis

 7. References should be updated and also provide the reference of the template protein

8. Type of interactions should be explained e.g., hydrogen bond, alkyl interaction etc 

Overall, in my opinion, the manuscript may be considered for publication after major corrections

Author Response

Please consider our revised manuscript ‘Homology modeling, molecular docking, molecular dynamic simulation,and drug-likeness of the modified alpha-mangostin against the β-tubulin protein of Acanthamoeba keratitis’. We would like to thank the reviewers for the time invested in reviewing this manuscript and for the valuable comments provided. We have considered the comments raised by the reviewers, and the responses detailed in the attached document have been implemented in the revised manuscript and highlighted with yellow colour.

Round 2

Reviewer 1 Report

Comments

1. Inform the version of the Gromacs package was used and the force field is GROMOS 43A1

Author Response

Submission of revised manuscript (Ref. No.: molecules-1911848)

Title: Homology modeling, molecular docking, molecular dynamic simulation,and drug-likeness of the modified alpha-mangostin against the β-tubulin protein of Acanthamoeba keratitis

Dear Dr. Dennis Zhou and Reviewer 1

Please consider our second revised manuscript ‘Homology modeling, molecular docking, molecular dynamic simulation, and drug-likeness of the modified alpha-mangostin against the β-tubulin protein of Acanthamoeba keratitis. We would like to thank Molecules and the reviewers for the time invested in reviewing this manuscript and for the valuable comments provided. We have considered the comments raised by the reviewers, and the responses detailed below have been implemented in the revised manuscript and highlighted with yellow colour.

Editor and Reviewer comments:

  1. Inform the version of the Gromacs package was used and the force field is GROMOS 43A1

Response

Thank you very much for the advice. I have added information about GROMOS96 43a1 forcefield and Gromacs package as follows:

The molecular dynamics were simulated for 50 ns by implying GROMOS96 43a1 forcefield with GROMACS (v5.1.2) package [66] using WebGro server.
